# Anesthesia for non-obstetric surgery during late term pregnancy in mares

**Pedro Henrique Salles Brito[1], Marília Alves Ferreira[1], Elidiane Rusch[1], Julia de Assis Arantes[1], Adriano Bonfim Carregaro[1], Carlos Augusto Araújo Valadão[2], Giovana Fumes Ghantous[3], Renata Gebara Sampaio Dória[1] ***

1 Department of Veterinary Medicine, Faculty of Animal Science and Food Engineering, University of São Paulo (USP), Pirassununga, São Paulo, Brazil, 2 Department of Veterinary Medicine and Surgery, Faculty of Agricultural and Veterinary Sciences, UNESP-São Paulo State University, Jaboticabal, São Paulo, Brazil, 3 Department of Basic Sciences, Faculty of Animal Science and Food Engineering, University of São Paulo (USP), Pirassununga, São Paulo, Brazil

☯ These authors contributed equally to this work.

* redoria@usp.br

**Data Availability Statement:** All relevant data are within the manuscript and its Supporting Information files.

## Abstract

Submitting late-term pregnant mares to anesthesia for non-obstetric surgery raises concerns about the survival of the mother and fetus. This study aimed to evaluate and describe transient maternal and fetal hemodynamic changes during general inhalation anesthesia in mares during the last month of gestation. Nine adult mares in the last month of gestation were subjected to general inhalation anesthesia and dorsal recumbency for 90 minutes. Trans-anesthetic vital parameters, arterial hemogasometry, cardiac output, pulmonary arterial pressure, central venous pressure, and fetal heart rates were assessed at defined intervals. During various timespans of the anesthetic procedure, the mares demonstrated an increase in heart rate, mean arterial pressure, and diastolic pressure as well as a decrease in temperature. Additionally, arterial hemogasometry indicated respiratory acidosis. No changes in cardiac output were observed; however, there was a reduction in pulmonary arterial and central venous pressures and stroke volume. Fetal heart rate was significantly decreased. General inhalation anesthesia in late term pregnancy in mares in a recumbent position implies in significant hemodynamic and metabolic changes. Up to 90 minutes those changes does not seem to affect negatively the maternal-fetus prognosis.

## Introduction

Subjecting pregnant mares to inhalation anesthesia often raises concerns regarding the mare and the fetus' survival, which can be of high economic or affective value. Clinicians and veterinary surgeons are usually asked about a foal's chance of survival after abdominal surgeries performed on their dam, and the answers are based mainly on the expertise of the surgeon and the anesthesiologist, as only a few studies involving this animal category are available [1–3]. The birth rate of viable foals after abdominal surgery ranges from 45 to 80% [3]. Multiple parameters are associated with maintaining pregnancy after anesthetic procedures, including duration of general anesthesia, hypotension, and intraoperative hypoxemia [2].

**Funding:** This research was funded by São Paulo Research Foundation (FAPESP) grants number [2020/09633-0 to CAAV, 2020/01387-0 to PHSB, and 2021/13378-8 to RGSD], and Technological Development (CNPq) grant number [309701/2022-8 to RGSD]. This study was financed in part by the Coordenação de Aperfeiçoamento de Pessoal de Nível Superior – Brasil (CAPES) – Finance Code 001. The funders had no role in study design, data collection and analysis, decision to publish, or preparation of the manuscript.

**Competing interests:** The authors have declared that no competing interests exist.

The most serious fetal risk associated with maternal anesthesia is intrauterine asphyxia. Since fetal oxygenation depends exclusively on maternal oxygenated blood, maintenance of maternal arterial oxygen pressure is essential, in addition to oxygen-carrying capacity and uteroplacental blood perfusion [4]. During general inhalation anesthesia of pregnant mares, there is a risk that maternal hemodynamic changes may compromise uteroplacental perfusion and/or fetal oxygenation, increasing the risk of abortion or preterm delivery. The increase in uterine volume observed in late-term mares associated with dorsal recumbency can cause compression of the caudal vena cava, which reduces venous return, preload, and cardiac output; culminating in systemic hypotension. In addition, compression of the abdominal aorta can reduce uterine blood flow [4, 5]. Thus, hemodynamic changes and changes in tissue oxygenation may alter the anesthetic risk in the last month of gestation for both the mother and the fetus.

The aim of this study was to perform trans-anesthetic maternal and fetal monitoring and describe their findings to assess the complexity and potential risks of general inhalation anesthesia in mares in the last month of gestation.

## Materials and methods

All procedures were approved by the Ethics Committee on Animal Use of Food Engineering and Animal Science, University of São Paulo, approval number 6532080921.

### Animals

Nine healthy mares in the last month of gestation (>300 days) were used. The animals were fasted from solids for 6 h before beginning the experiment.

### Experimental protocol

On the day of the experiment, the animals were placed in stocks. The left jugular vein was cannulated with a 14 gauge catheter for the administration of intravenous drugs. A Swan-Ganz thermodilution catheter (7.5 Fr, Edwards Lifesciences Corporation®, USA) was inserted into the right jugular vein. The ventrolateral abdomen was shaved from the sternum to the udder to facilitate fetal ultrasonographic assessment.

**General inhalation anesthesia and dorsal recumbency.** The animals received intravenous xylazine (1 mg kg$^{-1}$; Sedanew®, Vetnil, Brazil) as pre-anesthetic medication. Fifteen minutes later, anesthesia was induced with 10% ketamine (2.2 mg kg$^{-1}$; Cetamin®, Syntec, Brazil) and midazolam (0.1 mg kg$^{-1}$; Dormonid®, Roche, Brazil) administered intravenously in a single syringe. Following induction, the animals were hoisted by the legs onto the operating table in dorsal recumbency using an electric hoist. For maintenance of anesthesia, isoflurane (Isoforine®, Cristália, Brazil) diluted in 100% oxygen was delivered via an inhalation anesthesia machine equipped with a valve circuit (2800C, Mallard®, USA). End-tidal isoflurane concentration (EtISO) was adjusted to 2.25 V%, corresponding to a minimum alveolar concentration of 1.5, and controlled ventilation was adjusted up to 35 mmHg to maintain normocapnia (35–40 mmHg). The respiratory rate was adjusted to 10 breaths per minute, 1:2 inspiration/expiration ratio, and a tidal volume of 10 mL.kg$^{-1}$. Dobutamine solution (5 µg kg$^{-1}$min$^{-1}$; Dobutamine hydrochloride®, Laboratório Teuto, Brazil) was infused intravenously trying to maintain the minimum allowed arterial pressure (70 mmHg). General inhalation anesthesia was maintained for 90 min.

**Maternal trans-anesthetic monitoring.** Data collection started 15 min after the beginning of general inhalation anesthesia (T15), after fetal cardiac and ultrasonographic monitoring had already been established. The following parameters were measured every 10 min

during the trans-anesthetic period via a multiparameter monitor (LifeWindow[TM] LW9xVet®) Digicare Networked Multi-Parameter Veterinary Monitor): heart rate; invasive systolic, diastolic, and mean arterial blood pressure; oxygen saturation ($SpO_2$); and end-tidal carbon dioxide ($EtCO_2$).

Arterial blood gases were analyzed (i-stat Abbott®) in samples collected from the transverse facial artery immediately before pre-anesthetic medication (T0) and at T45, T75 and T90. The following parameters were evaluated: pH, arterial oxygen partial pressure ($PaO_2$), arterial carbon dioxide partial pressure ($PaCO_2$), total carbon dioxide ($tCO_2$), base excess, bicarbonate, $SO_2$, sodium, potassium, and calcium levels.

Cardiac output, pulmonary artery pressure, and central venous pressure were measured using a modified Swan-Ganz technique [6]. After the insertion of a Swan-Ganz thermodilution catheter (7.5 Fr, Edwards Lifesciences), a second port was placed in the right external jugular vein, distal to the catheter. Twelve human nasogastric tubes were inserted through this port into the right atrium and connected to a multiparameter monitor. Because the Swan-Ganz catheter is used exclusively in humans and because of the different dimensions of equine cardiac structures, it is necessary to adapt the technique to achieve correct measurement.

For the measurement of cardiac output, 0.1 mL kg$^{-1}$ of a 5% glucose solution at a temperature of 0 to 4˚C was administered through the nasogastric tube in a 5-second bolus. The same procedure was repeated sequentially until three values with a maximum difference of 0.5 liters were obtained. The mean value was multiplied by a correction factor of three to obtain the true cardiac output, which was measured in liters per minute from the multiparameter monitor with a constant set to 0.656. Central venous pressure was measured using the same tube connected to the multiparameter monitor. A Swan-Ganz catheter directly connected to a multiparameter monitor was used to measure the pulmonary artery pressure. The parameters were obtained at T0, T15, T25, T35, and T45, and every 15 min (T60, T75, and T90) until the end of the procedure. Cardiac output was used to determine the following parameters: cardiac index (cardiac output ÷ body weight), stroke volume (cardiac output ÷ heart rate), and peripheral vascular resistance ([mean arterial pressure–central venous pressure] ÷ cardiac output × 79.9).

**Fetal trans-anesthetic monitoring.** Fetal assessment was performed via transabdominal ultrasound (MyLab[TM] 30Gold, Esaote®) and measuring the heart rate at T0, T15, T25, T35, and T45, and every 15 minutes (T60, T75, and T90) until the end of the anesthetic procedure. After recovery from the anesthesia, further ultrasonographic assessment was performed.

After the anesthetic procedure, the mares were monitored and transabdominal ultrasonography was performed every day until the day of delivery. Births were assisted, and fetal viability was evaluated using the APGAR method modified for horses. Neonates underwent hematological and serum biochemistry analyses (renal function: urea and creatinine; liver function: AST, GGT, and alkaline phosphatase).

## Data analysis

Normality tests were performed for continuous quantitative variables when comparing between time points. For those that met the assumption of normality, a mixed model was adjusted; otherwise, gamma distribution was performed, and a generalized linear mixed model was used. For discrete quantitative variables, the Poisson distribution was used, and in cases of excess variance, the negative binomial model was used in the same structure used for the continuous variables. The procedures mixed and GENMOD from the SAS program (version 9.4; SAS Institute Inc.; Cary, NC, USA) were used. Additionally, Pearson correlations between the US Fetal variables and some variables of interest were evaluated using the Corr procedure of the program. The level of significance adopted for statistical tests was set at 5%.

## Results

The period of general inhalation anesthesia and dorsal recumbency of the late-term mares was standardized to 90 min because of difficulties in maintaining the hemodynamic and blood gas parameters of the mares and their fetuses within the physiological range of the species, which is necessary to ensure the safety of the anesthetic procedure for the animals.

The mares demonstrated an increase in heart rate, mean arterial pressure, and diastolic arterial pressure from T25 to T90 of the anesthetic procedure (T25, T35, T45, T55, T65, T75, T85, and T90) and a decrease in temperature from T55 to T90. Systolic arterial pressure increased at T75 (Figs 1–4; S1 Table).

Pulmonary ventilation was insufficient, with difficulty maintaining normocapnia, even with the peak pressure adjusted to 35 mmHg. The mean $PaO_2$ ranged from 89 ± 40.47 mmHg (T0) to 140.75 ± 56.45 mmHg (T75). The highest mean of $SpO_2$ was observed at T90 (98.16 ± 0.75%). The mean $PaCO_2$ remained between 50.51 ± 8.89 mmHg (T0) and 62.55 ± 5.07 mmHg (T75) and $EtCO_2$ between 38.89 ± 9.05 (T0) and 49.56 ± 9.59 (T55) during the trans-anesthetic period. There was an increase in PaO2 at T45 and T75, $PaCO_2$ at T75, and $SpO_2$ from T45 to T90 compared to those at T0. A reduction in arterial pH was observed at T75 and T90 (Figs 5–7). Mean sodium and potassium levels were reduced at T45, T75, and T90 but were within the physiological range for the species. Blood lactate remained above the physiological range from T0 to T90, without any difference between the time points (S2 Table).

No changes in cardiac output were observed; however, there was a reduction in pulmonary artery pressure from T15 to T90 (T15, T25, T35, T45, T60, T75, and T90) compared to that T0. Central venous pressure decreased from T45 to T75 (T45, T60, and T75; S3 Table) compared to that at T0. The stroke volume decreased at T25, T75, and T90 (Figs 8–11; S4 Table). No

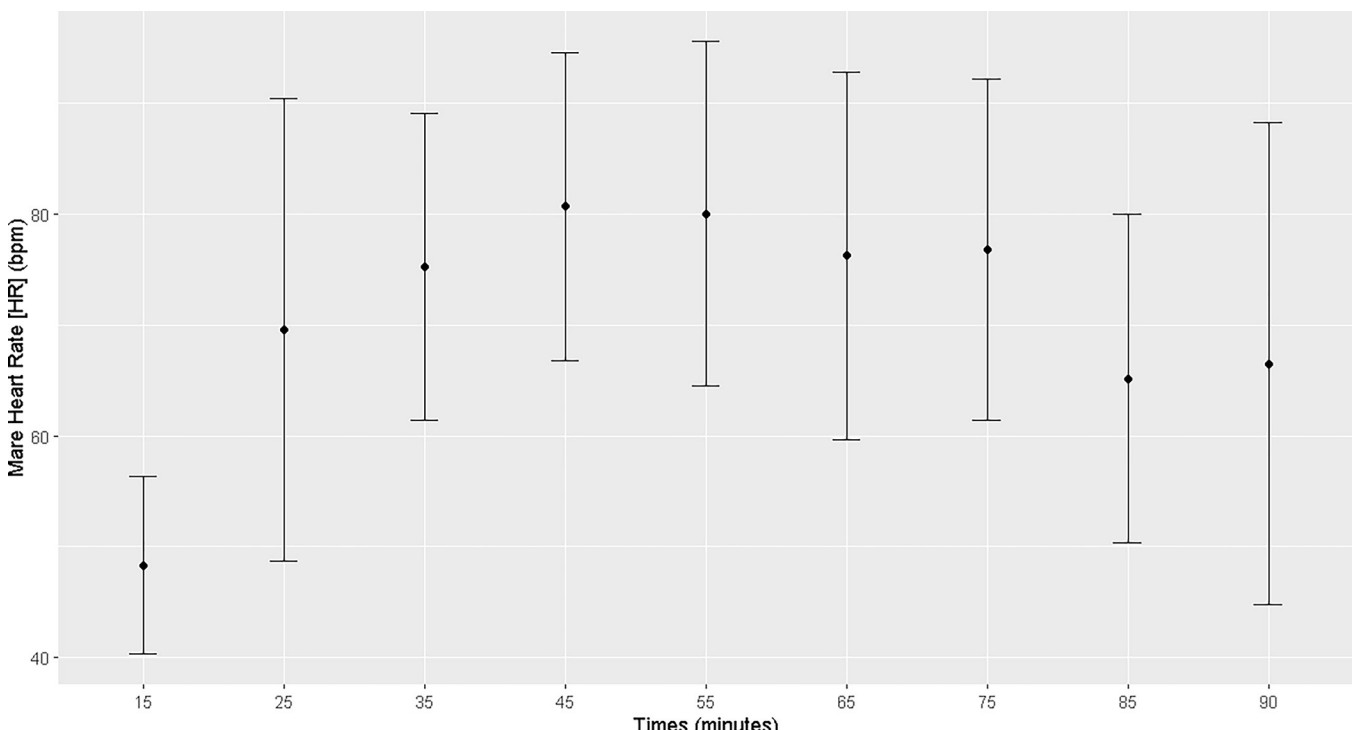

**Fig 1. Maternal heart rate.** Maternal heart rate (bpm) during general inhalation anesthesia and dorsal recumbency of mares in the last month of gestation.

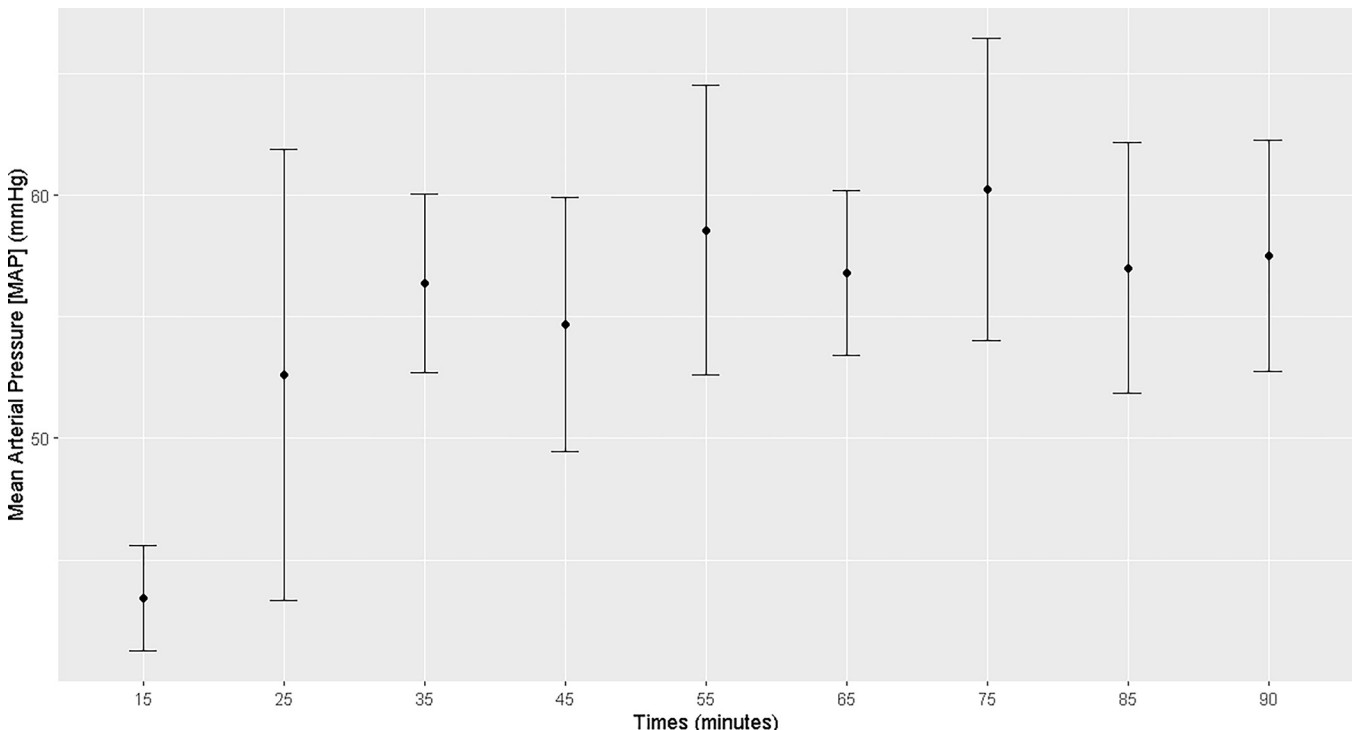

**Fig 2. Maternal mean arterial pressure.** Maternal mean arterial pressure (mmHg) during general inhalation anesthesia and dorsal recumbency of mares in the last month of gestation.

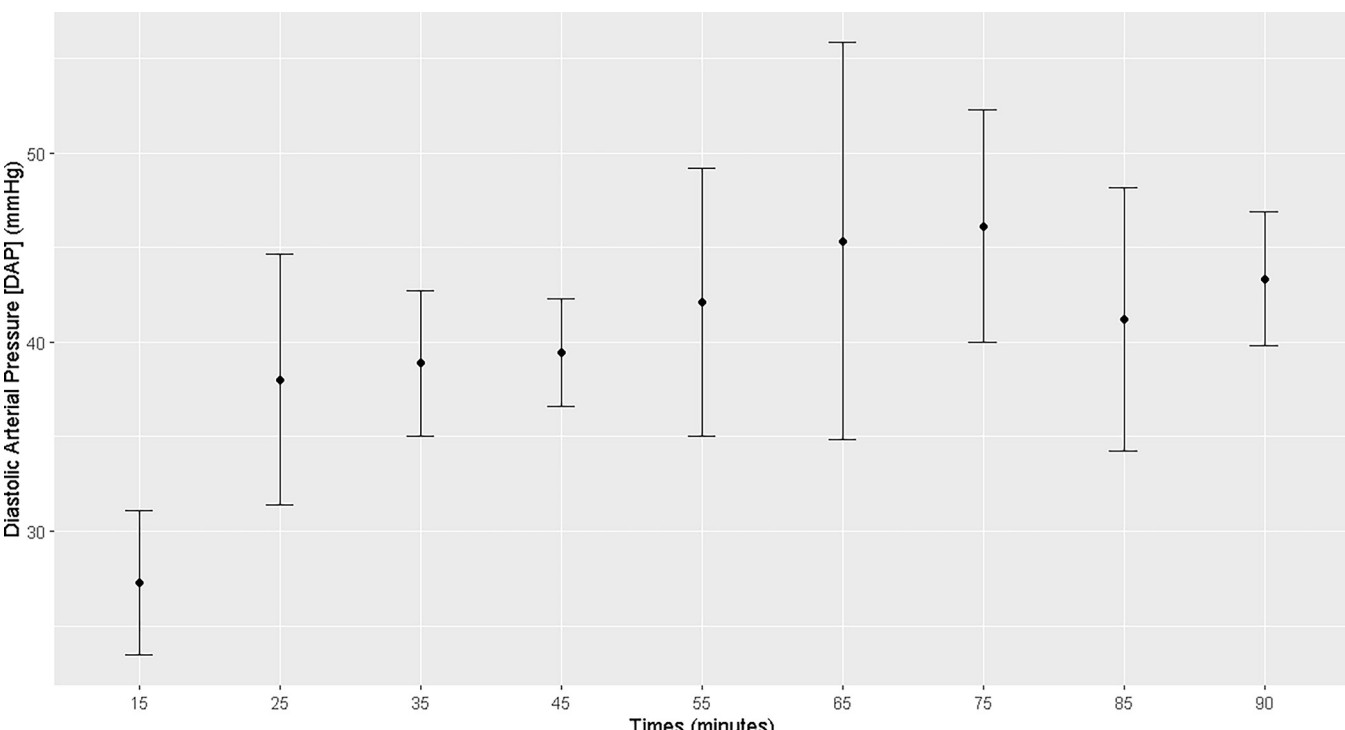

**Fig 3. Maternal diastolic arterial pressure.** Maternal diastolic arterial pressure during general inhalation anesthesia and dorsal recumbency of mares in the last month of gestation.

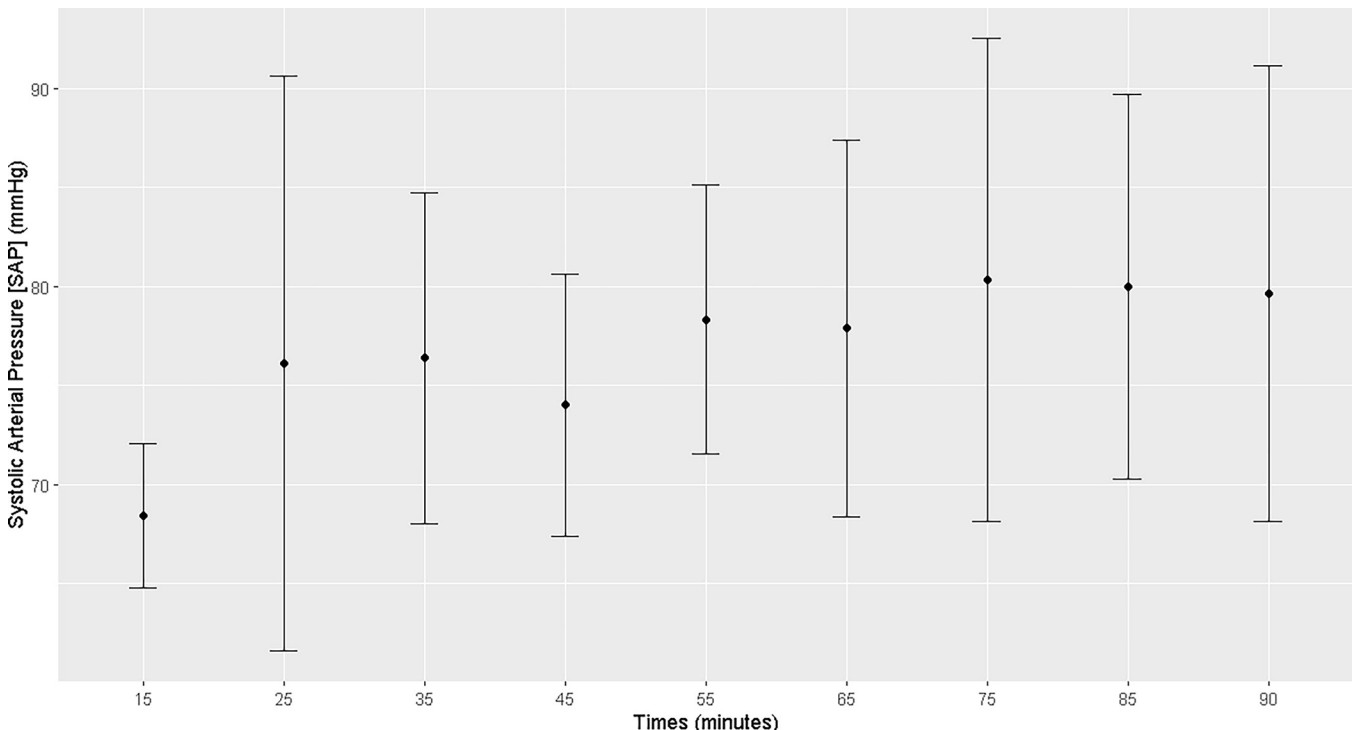

**Fig 4. Maternal systolic arterial pressure.** Maternal systolic arterial pressure (mmHg) during general inhalation anesthesia and dorsal recumbency of mares in the last month of gestation.

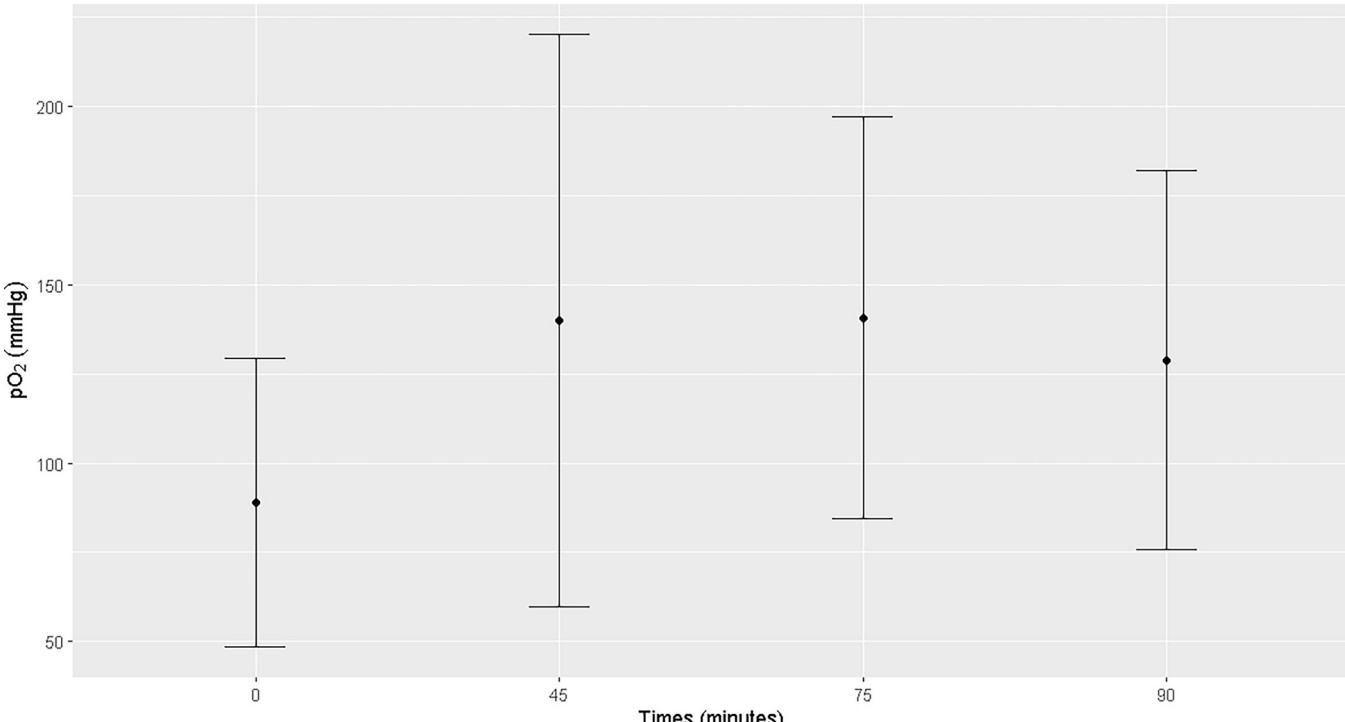

**Fig 5. Maternal blood pO2.** Maternal blood $pO_2$ (mmHg) at T0, T45, T75 and T90 of the general inhalation anesthesia and dorsal recumbency of mares in the last month of gestation.

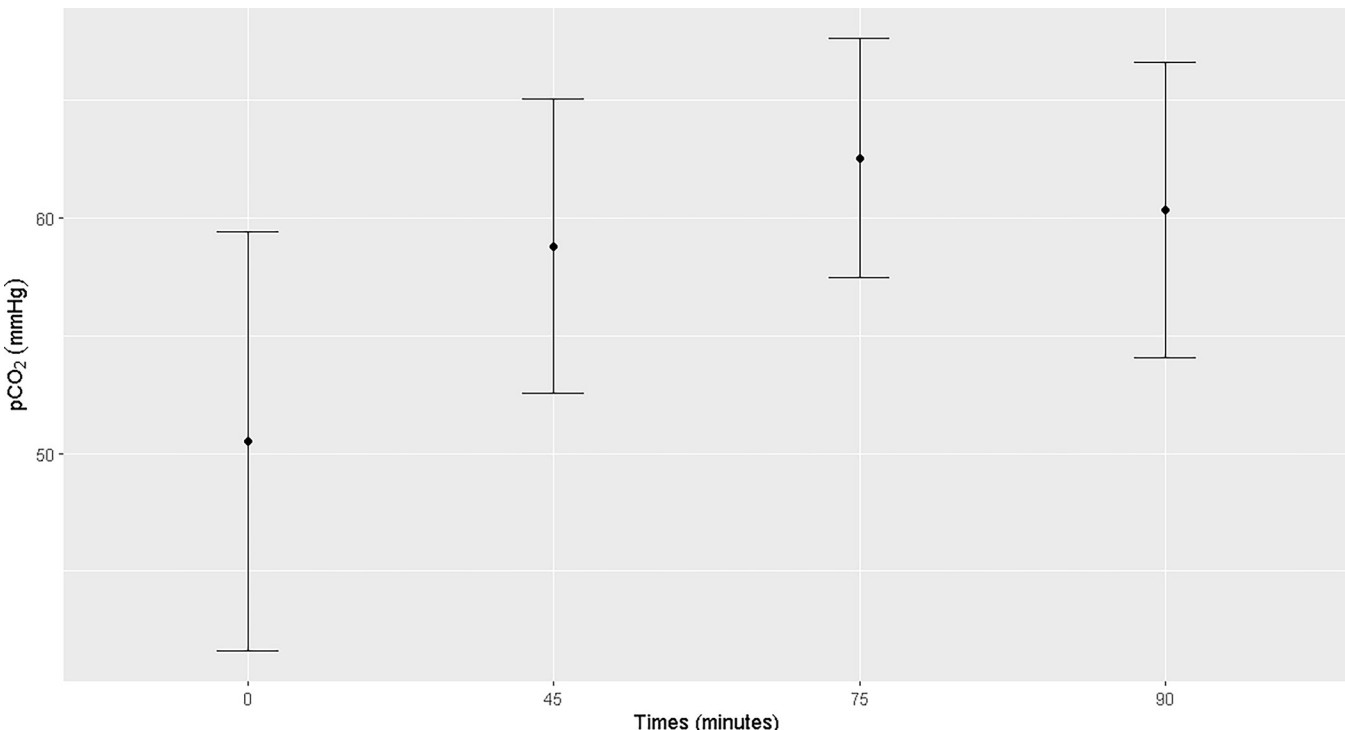

**Fig 6. Maternal blood pCO2 (mmHg).** Maternal blood $pCO_2$ (mmHg) at T0, T45, T75 and T90 of the general inhalation anesthesia and dorsal recumbency of mares in the last month of gestation.

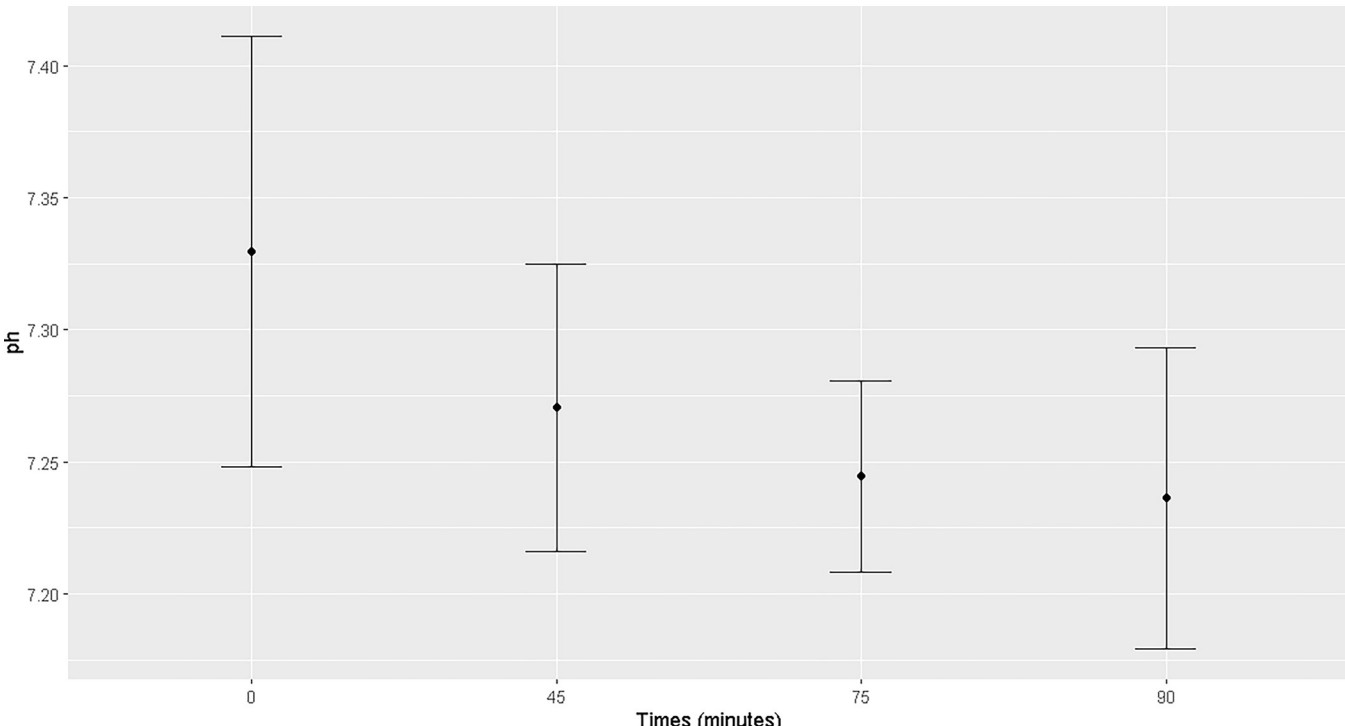

**Fig 7. Maternal blood pH.** Maternal blood pH at T0, T45, T75 and T90 of the general inhalation anesthesia and dorsal recumbency of mares in the last month of gestation.

changes over time were observed in mean cardiac index or peripheral vascular resistance (S4 Table).

In fetuses, a decrease in heart rate was observed starting at T25 and persisting up to T90, compared to that at T0 and T15. After recovery from anesthesia (Tpost), the fetal heart rate returned to mean values comparable to those observed at T0 (Fig 12; S5 Table).

A moderate positive correlation was observed between fetal heart rate and maternal arterial pH (0.55175; p = 0.0063) and a low negative correlation between fetal heart rate and maternal heart rate (-0.44215; p = 0.0038), mean arterial pressure (-0.33746; p = 0.0332), diastolic arterial pressure (-0.35062; p = 0.0265), and $PCO_2$ (-0.40083; p = 0.0580) (Table 1).

The deliveries occurred naturally without complications; 100% of the foals reached a maximum APGAR score of 8, and no alterations in hematological or serum biochemistry parameters were observed.

## Discussion

As far as we know, our paper is one of the first study to evaluate hemodynamic and blood gas changes in mares in the third trimester of pregnancy during general inhalation anesthesia and dorsal recumbency, and their effect on the fetus.

The mean $PaO_2$ values of mares, at T0, were below the expected levels for horses (89 ± 40.47 mmHg), possibly because of the increased intra-abdominal pressure exerted by the gravid uterus on the diaphragm, which reduces the amplitude of respiratory movements. After the induction of anesthesia, an increase in $PaO_2$ was observed in mares mechanically ventilated using 100% oxygen, although the mean values were close to the minimum recommended ones for anesthetic procedures (100–500 mmHg) [7]. Peak $PaO_2$ was observed at T45 and T75, with mean values of 139.88 ± 80.27 mmHg and 140.75 ± 56.45 mmHg, demonstrating difficulty in

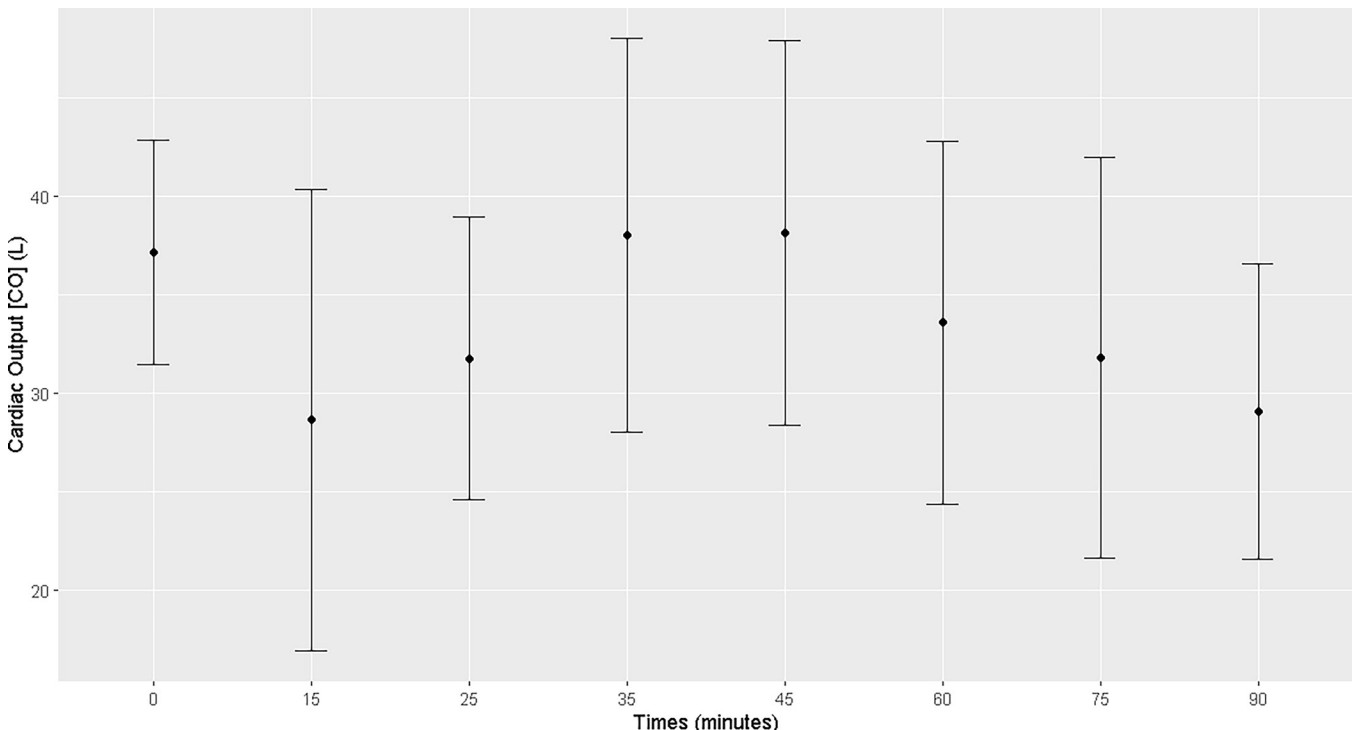

**Fig 8. Maternal cardiac output.** Maternal cardiac output (L) at T0 and during general inhalation anesthesia and dorsal recumbency of mares in the last month of gestation.

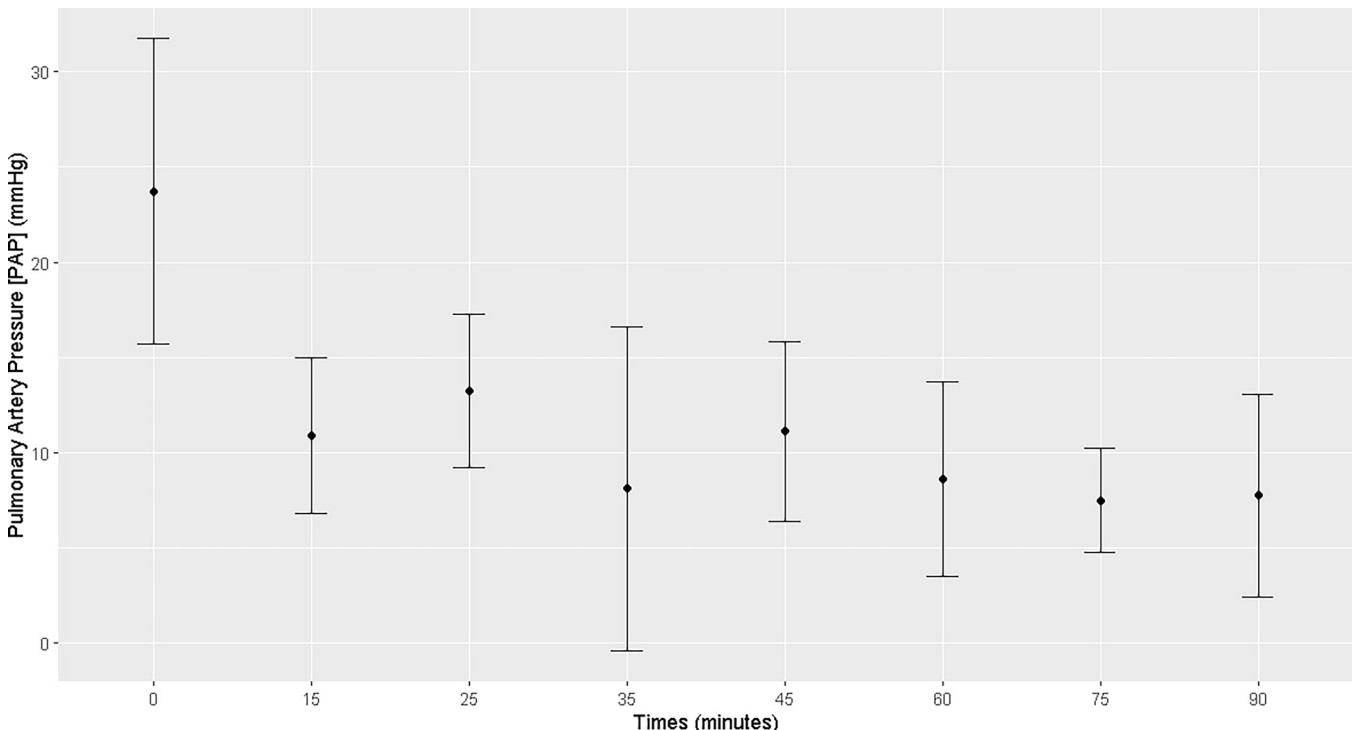

**Fig 9. Maternal pulmonary artery pressure.** Maternal pulmonary artery pressure (mmHg) at T0 and during general inhalation anesthesia and dorsal recumbency of mares in the last month of gestation.

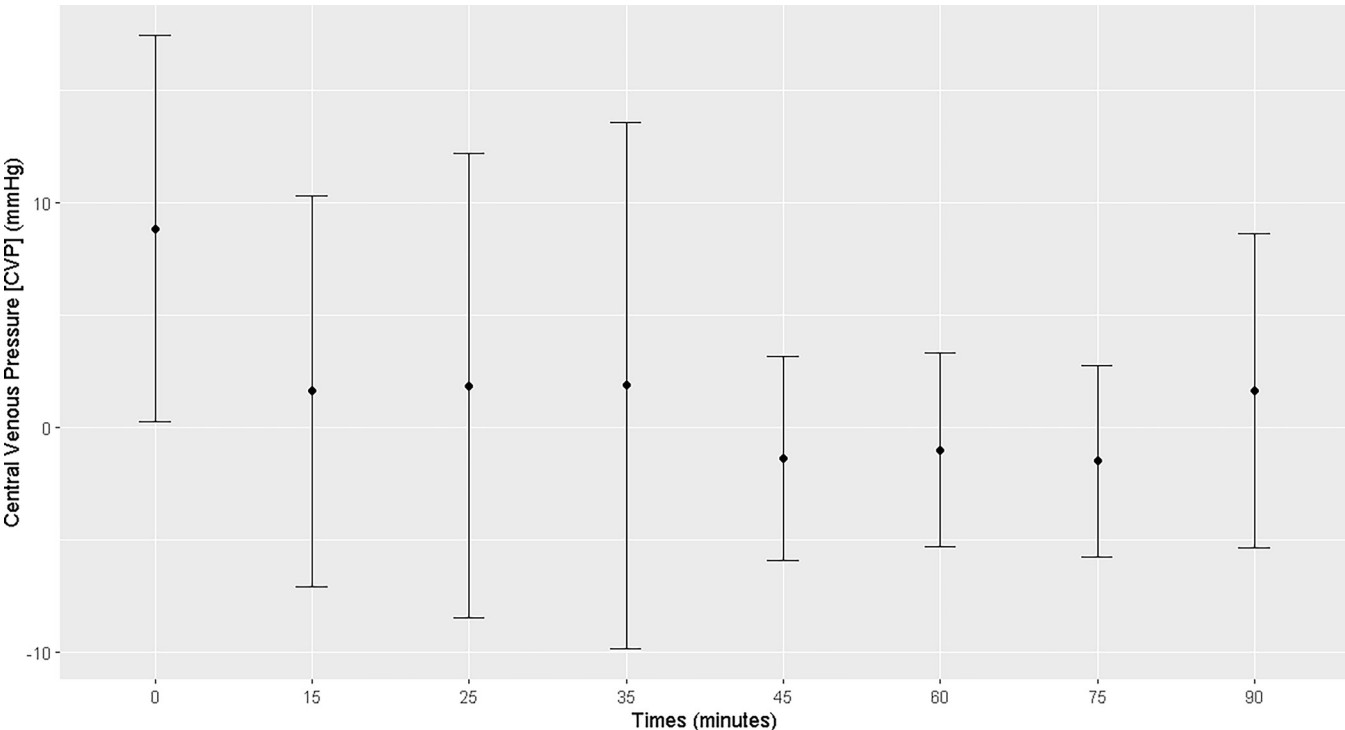

**Fig 10. Maternal central venous pressure.** Maternal central venous pressure (mmHg) at T0 and during general inhalation anesthesia and dorsal recumbency of mares in the last month of gestation.

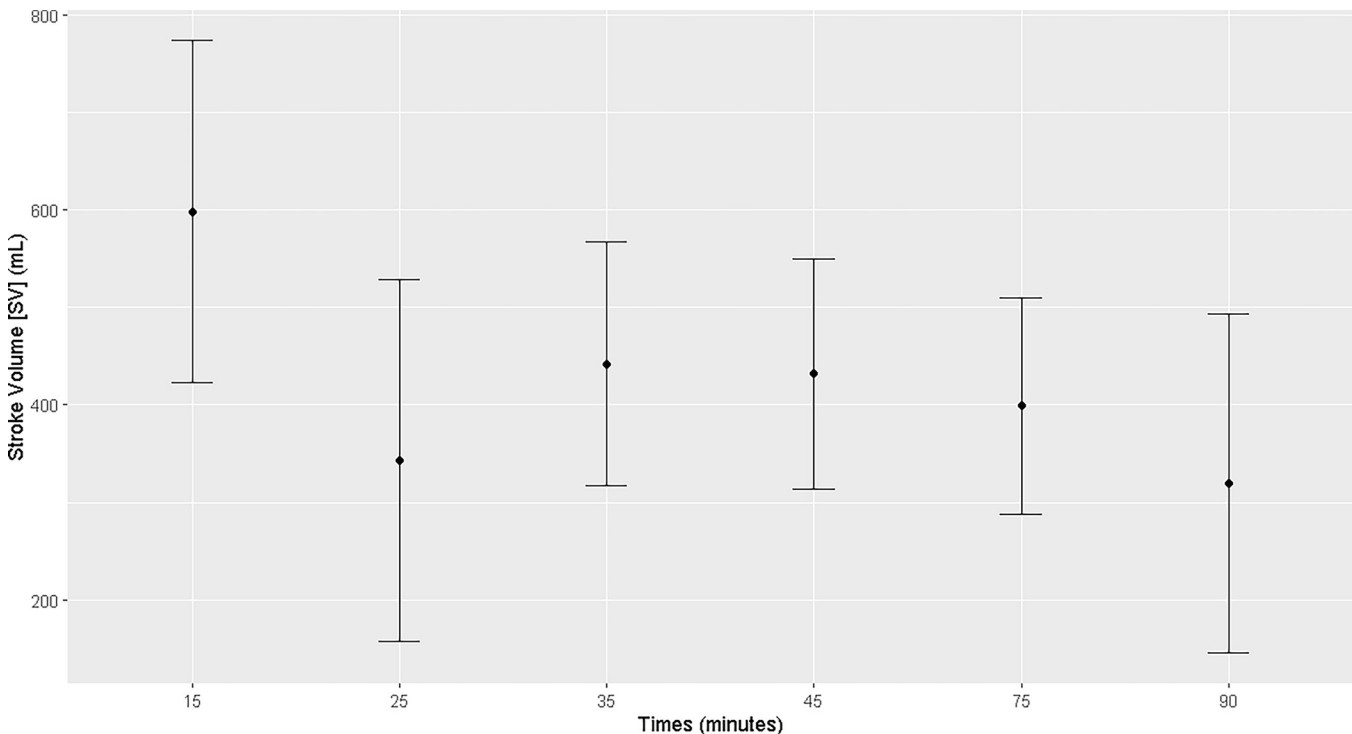

**Fig 11. Maternal stroke volume.** Maternal stroke volume (mL) during general inhalation anesthesia and dorsal recumbency of mares in the last month of gestation.

gas exchange due to the dorsal recumbent position and compression of the vena cava and the abdominal aorta caused by the gravid uterus [8].

$PaCO_2$ monitoring showed high mean values throughout the anesthesia, remaining close to the maximum recommended value for anesthetic procedures (40–60 mmHg), further demonstrating that gas exchange was inefficient [7]. The greater accumulation of carbon dioxide in blood was at T75, reaching a mean value of 62.55 ± 5.07 mmHg. Since hypercarbia is caused by hypoventilation [9], late term pregnant mares may be affected during general anesthesia and dorsal recumbency, as shown in this study. Hypercarbia [PaCO2 > 45 mmHg] and large alveolar-arterial oxygen differences are usually observed in horses during general anesthesia mainly caused by postural changes accompanying anesthesia, resulting in decreased functional residual capacity, ventilation-perfusion mismatching and atelectasis [10–12]. The development of atelectasis is significant in dorsally decumbent horses [13] and is observed in a significant number of horses despite normocarbia and increased $FiO_2$ [12, 14]. Hypercarbia in pregnant females can cause fetal acidosis because the $PaCO_2$ of the fetus is directly correlated with that of the mare; this can result in fetal cardiac depression and hypotension [8]. Physiologically, in humans, the $PaCO_2$ of pregnant women remains slightly lower (30 mmHg) than that of non-pregnant women, and maintenance of this hypocapnia is recommended during mechanical ventilation [15]. Thus comparatively, the increased $PaCO_2$ observed in late-term mares during general inhalation anesthesia and dorsal recumbency indicates a potential risk of fetal distress.

Starting at T0, the arterial pH of pregnant mares under general inhalation anesthesia was below the reference range for the species (7.37–7.49) [16], with a more marked reduction at T75 and T90. Despite the marked acidemia, bicarbonate levels remained unchanged during anesthesia. This indicates respiratory acidosis, as the rise in $PaCO_2$ caused the decrease in

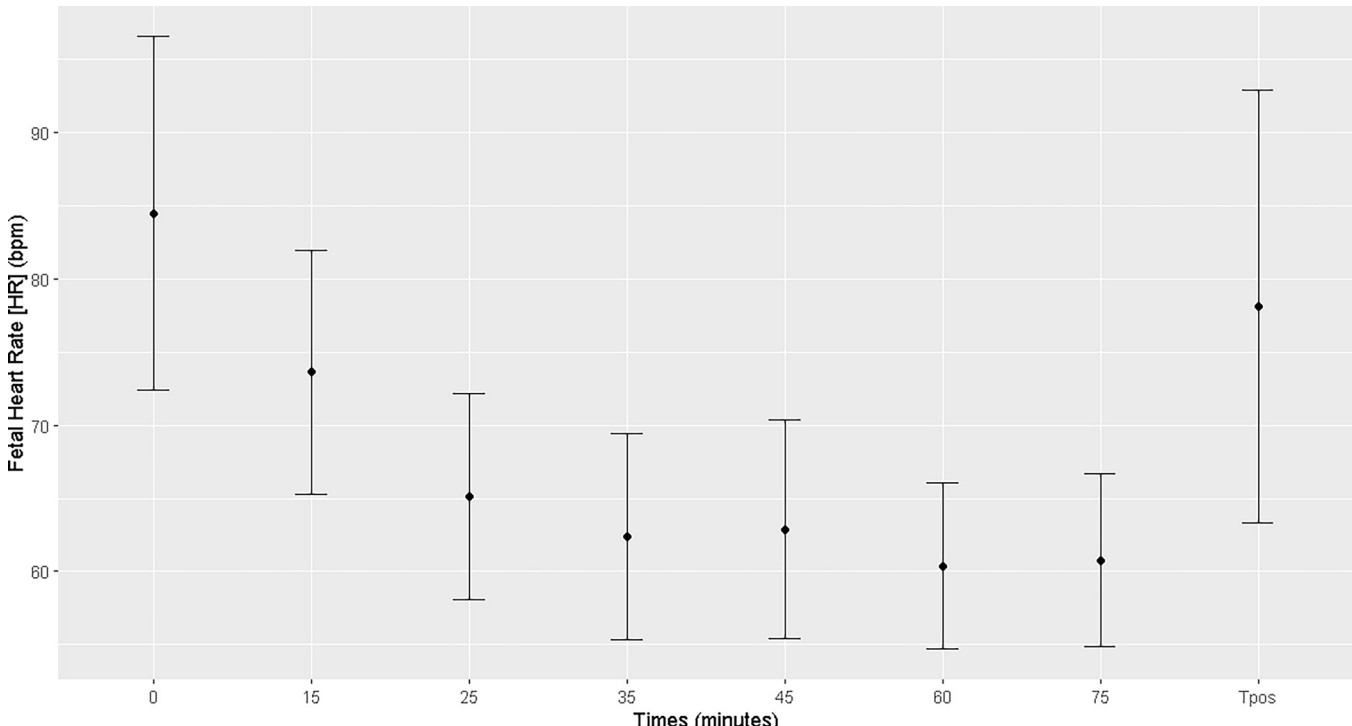

**Fig 12. Fetal heart rate.** Fetal heart rate (bpm) at T0, during general inhalation anesthesia and dorsal recumbency of mares in the last month of gestation, and after maternal recovery from anesthesia.

blood pH [7]. There was also an inversion of the base deficit from positive values at T0 and T45 to negative values at T75 and T90, with mean values below the physiological range for the species (-0.51 to 8.80 mmol/L) [16] at T90 (-1.75 ± 4.56). This inversion of values indicates a

**Table 1. Pearson correlation of maternal variables and fetal heart rate.**

| Variables | N | Pearson Correlation | p-value (p< 0.05) |
|---|---|---|---|
| HR (bpm) | 41 | -0.44215 | 0.0038 |
| SAP (mmHg) | 40 | -0.12781 | 0.4319 |
| MAP (mmHg) | 40 | -0.33746 | 0.0332 |
| DAP (mmHg) | 40 | -0.35062 | 0.0265 |
| pH | 23 | 0.55175 | 0.0063 |
| paCO$_2$ (mmHg) | 23 | -0.40083 | 0.0580 |
| paO$_2$ (mmHg) | 23 | -0.26038 | 0.2302 |
| Lactate (mmol/L) | 23 | 0.14011 | 0.5237 |
| PAP (mmHg) | 54 | 0.28533 | 0.0365 |
| CVP (mmHg) | 53 | 0.13748 | 0.3262 |
| CO (L) | 54 | -0.09726 | 0.4841 |
| CI (mL) | 54 | -0.12167 | 0.3808 |
| SV (ml/beats) | 40 | 0.09369 | 0.5653 |
| PVR | 38 | 0.16705 | 0.3161 |

Pearson correlation of maternal variables (heart rate–HR, systolic arterial pressure–SAP, mean arterial pressure–MAP, diastolic arterial pressure–DAP, pH, paCO$_2$, paO$_2$, lactate, pulmonary arterial pressure–PAP, central venous pressure–CVP, cardiac output–CO, cardiac index–CI, stroke volume–SV and peripheral vascular resistance–PVR) and fetal heart rate (bpm) during general inhalation anesthesia of mares in the last month of gestation.

deficiency of bases [17] and is directly related to a reduction in the mean blood pH without the physiological compensation via bicarbonates. At the same time points, the mean $PaCO_2$ values were above the reference range for the species (> 60 mmHg) [7].

Maternal heart rate was elevated throughout the anesthetic procedure, which is associated with the low $PaO_2$ and high $PaCO_2$ levels and the acidemia. Hypotension was observed analogously. Under continuous infusion of dobutamine (5 µg kg$^{-1}$min$^{-1}$), both the mean arterial pressure and diastolic blood pressure increased starting at T25, but remained below the physiological range [7]. It was not possible to maintain a mean arterial pressure > 60 mmHg. Uteroplacental perfusion is strongly correlated with maternal blood pressure; thus, low maternal pressure can result in fetal asphyxia. In humans, blood pressures higher than 70–80 mmHg ensures good uterine perfusion [18]. In the present study, maintaining dobutamine infusion was necessary throughout the anesthetic procedure to sustain good tissue perfusion and, consequently, uterine perfusion [19]. However, hypotension remained evident in the anesthetized mares, posing a potential risk of fetal distress.

The compression exerted by the gravid uterus and its appendages on the caudal vena cava causes a significant reduction in the blood volume entering the right atrium (cardiac preload), thus compromising diastolic filling and cardiac pumping capacity. This, in turn, decreases the diastolic blood pressure and consequently the mean arterial pressure [8, 20].

In the literature consulted, there are no reference values for studying the hemodynamics of mares in the third trimester of pregnancy submitted to general inhalation anesthesia and dorsal recumbency. In our study, cardiac output remained unchanged throughout the anesthetic procedure despite the hypotension shown by the animals, demonstrating that the bodies of pregnant mares were able to efficiently maintain blood flow. Although the stroke volume of the animals decreased during the anesthetic procedure and presented values lower than those reported in the literature, the increase in heart rate was probably responsible for keeping the cardiac output constant and comparable to the values previously reported, which are considered the physiological range for horses [7, 21]. The cardiac index was calculated to standardize the cardiac output as a function in the animals of varying weights and sizes, thus permitting an adjusted and reliable correlation of the data [22]. In the present study, the cardiac indices and vascular peripheral resistance remained constant over time within the physiological range of the species, corroborating the cardiac output results and confirming the effectiveness of the physiological response of pregnant mares to hemodynamic changes.

Before beginning the anesthetic procedure, the central venous pressure and pulmonary arterial pressure of pregnant mares were within the expected range for this species (5.9–9 mmHg; 22–28 mmHg respectively) [21, 23]. After the induction of anesthesia and placing the animals in dorsal recumbency, the central venous pressure and pulmonary arterial pressure decreased to levels below the expected values. The median central venous pressure was negative at T45, T60, and T75, demonstrating that the preload was negatively influenced. Some of the factors influencing central venous and pulmonary arterial pressures include the use of positive pressure during mechanical ventilation, cardiac tamponade, pleural effusion, and increased intra-abdominal pressure [24]. In this case, in addition to positive pressure mechanical ventilation, the gravid uterus pressing on the caudal vena cava may have also been a cause of the low central venous pressure since the beginning of the anesthetic procedure [8]. Similarly, the decreased pulmonary arterial pressure from T15 to T90 may be explained by anesthesia and recumbency [21].

Despite the low stroke volume resulting from impaired venous return and low, sometimes even negative, central venous pressure both caused by the uterus compressing the abdominal vena cava; there was a compensatory increase in the maternal heart rate aimed at maintaining the cardiac output within the necessary limits to ensure good tissue perfusion and, consequently, uterine perfusion.

In addition to maternal findings, this study identified numerous factors that increase the risk of fetal distress during general inhalation anesthesia and dorsal recumbency of mares in the last month of gestation. The fetal bradycardia observed during the maternal procedure in the present study can be explained by the combined effect of anesthetic drugs and probable fetal hypoxia caused by reduction in maternal blood supply to the fetus [8]; the latter is caused by tissue hypoperfusion due to hypotension and compromised maternal gas exchange [25]. A comparison of maternal tans-anesthetic heart rate monitoring with that of the fetus revealed a reduction in fetal heart rate by T25, which continued until the end of general inhalation anesthesia (T90). Fetal heart rate presented values nearer to and lower than the minimum accepted as physiologic in this phase of pregnancy (105±4 to 79±3 bpm) [26]. At that point, it was deemed necessary to interrupt the anesthetic procedure because the maternal hemodynamic and blood gas changes varied widely, and the mean values were outside the range of safe anesthesia. The reduced maternal blood pH and elevated $PaCO_2$ at a mean arterial pressure below the safe range for the species compromised gas exchange in tissues, including in the gravid uterus, directly affecting fetal hemodynamics and, subsequently, causing fetal distress.

This study showed that heart rate, blood pH, and $PaCO_2$ are correlated with fetal heart rate. To summarize, as maternal heart rate and $PaCO_2$ increase, the blood pH decreases, resulting in reduction in fetal heart rate. These data could pose an anesthetic risk for the fetus. For human surgeons, low trans-anesthetic fetal activity associated with a pronounced decrease in fetal heart rate during prolonged surgical procedures can raise concerns regarding the need for cesarean section, especially in patients at advanced gestational ages, considering aspects related to fetal suffering [15].

In the present study, all nine foals were born at term by natural delivery without complications in mares or foals. Thus, considering the possible economic costs and harmful health effects associated with the birth of a premature or dysmature foal [27], obstetric interventions such as cesarean section are not recommended unless there is an imminent risk of maternal death. Although signs of fetal distress were observed during the anesthetic period, they were transient and controllable with good maternal anesthetic monitoring. The return of fetal heart rate to physiological levels and the restoration of fetal activity were observed immediately after recovery from anesthesia, and there were no hematological changes or alterations in the liver or kidney functions.

The main limitation of the present study was the lack of comparisons with a control group. The second limitation was the small number of mares used, a fact that increases the individual variation. The third limitation was that general inhalation anesthesia and dorsal recumbency were not evaluated in mares at other stages of pregnancy.

## Conclusion

The present study indicates that general inhalation anesthesia in late term pregnancy in mares in a recumbent position implies in significant hemodynamic and metabolic changes. Up to 90 minutes those changes does not seem to affect negatively the maternal-fetus prognosis. However, further studies should be conducted, with bigger sample sizes and comparison with controls and other phases of pregnancy in the mare, to support this conclusion. Our findings may help improve the surgical management of late-term pregnant mares.

## Supporting information

**S1 Table. General inhalation anesthesia parameters of mares in the last month of gestation.** Mean and standard deviation of temperature (˚C), respiratory rate (rpm), heart rate (bpm), end-tidal carbon dioxide ($EtCO_2$), oxygen saturation ($SpO_2$), systolic (SAP; mmHg),

mean (MAP; mmHg) and diastolic (DAP; mmHg) arterial pressure during general inhalation anesthesia of mares in the last month of gestation.
(DOCX)

**S2 Table. General inhalation anesthesia hemogasometry of mares in the last month of gestation.** Mean and standard deviation of pH, $pCO_2$ (mmHg), $pO_2$ (mmHg), base excess (mmol/L), $HCO_3$ (mmol/L), $tCO_2$ (mmol/L), $SO_2$ (%), sodium (mmol/L), potassium (mmol/L), calcium (mmol/L), and lactate (mmol/L) during general inhalation anesthesia of mares in the last month of gestation.
(DOCX)

**S3 Table. General inhalation anesthesia hemodynamic of mares in the last month of gestation.** Mean and standard deviation of pulmonary arterial pressure (PAP; mmHg), central venous pressure (CVP; mmHg), cardiac output (CO; L), and cardiac index (CI; mL) during general inhalation anesthesia of mares in the last month of gestation.
(DOCX)

**S4 Table. General inhalation anesthesia hemodynamic of mares in the last month of gestation.** Mean and standard deviation of stroke volume (SV; mL/beats) and peripheral vascular resistance (PVR) during general inhalation anesthesia of mares in the last month of gestation.
(DOCX)

**S5 Table. Fetal heart rate.** Mean and standard deviation of fetal heart rate (bpm) during general inhalation anesthesia of mares in the last month of gestation.
(DOCX)

**S6 Table. Raw data.** Maternal heart rate. Maternal heart rate (bpm) during general inhalation anesthesia and dorsal recumbency of mares in the last month of gestation.
(DOCX)

**S7 Table. Raw data.** Maternal respiratory rate. Maternal respiratory rate (rpm) during general inhalation anesthesia and dorsal recumbency of mares in the last month of gestation.
(DOCX)

**S8 Table. Raw data.** Maternal mean arterial pressure. Maternal mean arterial pressure (mmHg) during general inhalation anesthesia and dorsal recumbency of mares in the last month of gestation.
(DOCX)

**S9 Table. Raw data.** Maternal diastolic arterial pressure. Maternal diastolic arterial pressure (mmHg) during general inhalation anesthesia and dorsal recumbency of mares in the last month of gestation.
(DOCX)

**S10 Table. Raw data.** Maternal systolic arterial pressure. Maternal systolic arterial pressure (mmHg) during general inhalation anesthesia and dorsal recumbency of mares in the last month of gestation.
(DOCX)

**S11 Table. Raw data.** Maternal temperature. Maternal temperature (˚C) during general inhalation anesthesia and dorsal recumbency of mares in the last month of gestation.
(DOCX)

**S12 Table. Raw data.** Maternal $pO_2$. Maternal $pO_2$ (mmHg) during general inhalation anesthesia and dorsal recumbency of mares in the last month of gestation.
(DOCX)

**S13 Table. Raw data.** Maternal $pCO_2$. Maternal $pCO_2$ (mmHg) during general inhalation anesthesia and dorsal recumbency of mares in the last month of gestation.
(DOCX)

**S14 Table. Raw data.** Maternal pH. Maternal pH during general inhalation anesthesia and dorsal recumbency of mares in the last month of gestation.
(DOCX)

**S15 Table. Raw data.** Maternal Cardiac Output. Maternal cardiac output (L) during general inhalation anesthesia and dorsal recumbency of mares in the last month of gestation.
(DOCX)

**S16 Table. Raw data.** Maternal Pulmonary Artery Pressure. Maternal pulmonary artery pressure (mmHg) during general inhalation anesthesia and dorsal recumbency of mares in the last month of gestation.
(DOCX)

**S17 Table. Raw data.** Maternal Central Venous Pressure. Maternal central venous pressure (mmHg) during general inhalation anesthesia and dorsal recumbency of mares in the last month of gestation.
(DOCX)

**S18 Table. Raw data.** Maternal Stroke Volume. Maternal stroke volume (mL) during general inhalation anesthesia and dorsal recumbency of mares in the last month of gestation.
(DOCX)

**S19 Table. Raw data.** Fetal heart rate. Fetal heart rate (bpm) at T0, during general inhalation anesthesia and dorsal recumbency of mares in the last month of gestation, and after maternal recovery from anesthesia.
(DOCX)

**S20 Table. Raw data.** Maternal $EtCO_2$. Maternal $EtCO_2$ during general inhalation anesthesia and dorsal recumbency of mares in the last month of gestation.
(DOCX)

**S21 Table. Raw data.** Maternal $SpO_2$. Maternal $SpO_2$ during general inhalation anesthesia and dorsal recumbency of mares in the last month of gestation.
(DOCX)

**S22 Table. Raw data.** Maternal Base Excess. Maternal base excess (mmol/L) during general inhalation anesthesia and dorsal recumbency of mares in the last month of gestation.
(DOCX)

**S23 Table. Raw data.** Maternal $HCO_3$. Maternal $HCO_3$ during general inhalation anesthesia and dorsal recumbency of mares in the last month of gestation.
(DOCX)

**S24 Table. Raw data.** Maternal $tCO_2$. Maternal $tCO_2$ during general inhalation anesthesia and dorsal recumbency of mares in the last month of gestation.
(DOCX)

**S25 Table. Raw data.** Maternal $SO_2$. Maternal $SO_2$ during general inhalation anesthesia and dorsal recumbency of mares in the last month of gestation.
(DOCX)

**S26 Table. Raw data.** Maternal Sodium. Maternal sodium (mmol/L) during general inhalation anesthesia and dorsal recumbency of mares in the last month of gestation.
(DOCX)

**S27 Table. Raw data.** Maternal Potassium. Maternal potassium (mmol/L) during general inhalation anesthesia and dorsal recumbency of mares in the last month of gestation.
(DOCX)

**S28 Table. Raw data.** Maternal Calcium. Maternal calcium (mmol/L) during general inhalation anesthesia and dorsal recumbency of mares in the last month of gestation.
(DOCX)

**S29 Table. Raw data.** Maternal Lactate. Maternal lactate (mmol/L) during general inhalation anesthesia and dorsal recumbency of mares in the last month of gestation.
(DOCX)

**S30 Table. Raw data.** Maternal Cardiac Index. Maternal cardiac index (mL) during general inhalation anesthesia and dorsal recumbency of mares in the last month of gestation.
(DOCX)

**S31 Table. Raw data.** Maternal Peripheric Vascular Resistance. Maternal peripheric vascular resistance during general inhalation anesthesia and dorsal recumbency of mares in the last month of gestation.
(DOCX)

## Author Contributions

**Conceptualization:** Pedro Henrique Salles Brito, Marília Alves Ferreira, Adriano Bonfim Carregaro, Carlos Augusto Araújo Valadão, Giovana Fumes Ghantous, Renata Gebara Sampaio Dória.

**Data curation:** Pedro Henrique Salles Brito, Giovana Fumes Ghantous, Renata Gebara Sampaio Dória.

**Formal analysis:** Pedro Henrique Salles Brito, Giovana Fumes Ghantous, Renata Gebara Sampaio Dória.

**Funding acquisition:** Pedro Henrique Salles Brito, Renata Gebara Sampaio Dória.

**Investigation:** Pedro Henrique Salles Brito, Marília Alves Ferreira, Elidiane Rusch, Julia de Assis Arantes, Adriano Bonfim Carregaro, Carlos Augusto Araújo Valadão, Giovana Fumes Ghantous, Renata Gebara Sampaio Dória.

**Methodology:** Pedro Henrique Salles Brito, Marília Alves Ferreira, Elidiane Rusch, Julia de Assis Arantes, Adriano Bonfim Carregaro, Renata Gebara Sampaio Dória.

**Project administration:** Pedro Henrique Salles Brito, Renata Gebara Sampaio Dória.

**Resources:** Pedro Henrique Salles Brito, Marília Alves Ferreira, Elidiane Rusch, Julia de Assis Arantes, Adriano Bonfim Carregaro, Carlos Augusto Araújo Valadão, Giovana Fumes Ghantous, Renata Gebara Sampaio Dória.

**Software:** Giovana Fumes Ghantous, Renata Gebara Sampaio Dória.

**Supervision:** Adriano Bonfim Carregaro, Carlos Augusto Araújo Valadão, Renata Gebara Sampaio Dória.

**Validation:** Carlos Augusto Araújo Valadão, Renata Gebara Sampaio Dória.

**Visualization:** Carlos Augusto Araújo Valadão, Renata Gebara Sampaio Dória.

**Writing – original draft:** Pedro Henrique Salles Brito, Giovana Fumes Ghantous, Renata Gebara Sampaio Dória.

**Writing – review & editing:** Pedro Henrique Salles Brito, Marília Alves Ferreira, Elidiane Rusch, Julia de Assis Arantes, Adriano Bonfim Carregaro, Carlos Augusto Araújo Valadão, Giovana Fumes Ghantous, Renata Gebara Sampaio Dória.

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
