## [Decision Letter · Decision Letter 0]

7 Aug 2024

PONE-D-24-15168Anesthesia for non-obstetric surgery during late term pregnancy in maresPLOS ONE

Dear Dr. Dória,

Thank you for submitting your manuscript to PLOS ONE. After careful consideration, we feel that it has merit but does not fully meet PLOS ONE’s publication criteria as it currently stands. Therefore, we invite you to submit a revised version of the manuscript that addresses the points raised during the review process.

We look forward to receiving your revised manuscript.

Kind regards,

Daniel Negrini Medeiros, PhD

Guest Editor

PLOS ONE

Journal Requirements:

"This research was funded by São Paulo Research Foundation (FAPESP) grants number [2020/01387-0 PHSB and 2023/13378-8 RGSD], and Technological Development (CNPq) grant number [309701/2022-8 RGSD].  And the APC was funded by [FAPESP; grants number 2020/01387-0 PHSB and 2023/13378-8 RGSD]. This study was financed in part by the Coordenação de Aperfeiçoamento de Pessoal de Nível Superior – Brasil (CAPES) – Finance Code 001."

**Additional Editor Comments:**

I believe the paper has merits. However, to reach level for publication in such a prestigious journal as PLOS ONE, the authors should take carefull consideration on the comments made by both reviewers. Additionally, authors should be more cautious only to conclude what has be supported by their results. Also, all authors show us, in the results section, a descriptive statistic, which is ok. However, they should not state they have made comparisons. By the way, the lack of a control group should the highlighted as a importante limitation of their work.

Reviewers' comments:

Reviewer's Responses to Questions

**Comments to the Author**

1. Is the manuscript technically sound, and do the data support the conclusions?

Reviewer #1: Partly

Reviewer #2: Partly

2. Has the statistical analysis been performed appropriately and rigorously? 

Reviewer #1: Yes

Reviewer #2: No

3. Have the authors made all data underlying the findings in their manuscript fully available?

Reviewer #1: Yes

Reviewer #2: Yes

4. Is the manuscript presented in an intelligible fashion and written in standard English?

Reviewer #1: Yes

Reviewer #2: Yes

5. Review Comments to the Author

Reviewer #1: I believe al my relevant comments are already in the attached document. The data presented by the authors are interesting. However I believe hey haven't highlighted the most relevant data. Moreover, the discussion should be revised in a way that it reflects what is shown at the results section. Additionally, the results also does not support the authors conclusions, and limitations should also be revised.

Reviewer #2: Statistic analysis should be clearly described in the revised manuscript. The authors stated that they had performed comparisons. However, they only reported linear correlation coefficients. Conclusions did not derived from the statistical significance of the data. Moreover, correction for multiple comparisons were not described.

6. PLOS authors have the option to publish the peer review history of their article (what does this mean?). If published, this will include your full peer review and any attached files.

Reviewer #1: **Yes: **Tatiana Plais de Castro Linhrares

Reviewer #2: No

---

## [Author Response · Author response to Decision Letter 0]

9 Sep 2024

Dear editor and reviewers,

Thank you very much for the attention and the chance of publishing an article in so important journal. We took care of all the comments and suggestions that the reviewers had done carefully and we made all the changes suggested and respond all the comments individually. All the changes we have done in the manuscript are highlighted (yellow).

So, we hope the preparation of this revision will permit a positive final decision on my report's suitability for publication in Plos One.

Thank you very much for the additional comments.

Best regards.

Renata G S Dória et al.

1. We have ensured that our manuscript meets PLOS ONE's style requirements, as suggested. 

2. We need to include this paragraph, as suggested. Please include this amended for us.

This research was funded by São Paulo Research Foundation (FAPESP) grants number [2020/09633-0 CAAV, 2020/01387-0 PHSB, and 2021/13378-8 RGSD], and Technological Development (CNPq) grant number [309701/2022-8 RGSD]. And the APC was funded by [FAPESP; grants number 2020/09633-0 CAAV, 2020/01387-0 PHSB, and 2021/13378-8 RGSD]. This study was financed in part by the Coordenação de Aperfeiçoamento de Pessoal de Nível Superior – Brasil (CAPES) – Finance Code 001. The funders had no role in study design, data collection and analysis, decision to publish, or preparation of the manuscript. 

3. We have included captions for our Supporting Information files at the end of our manuscript, and update the in-text citations to match accordingly. 

Dear Editor,

We have taken careful consideration on the comments made by both reviewers. Additionally, we were more cautious to conclude what can be supported by our results, as suggested. 

We removed the term “comparison” in the text, as suggested. And we added the lack of a control group as an important limitation of our work.

Reviewer #1: I believe all my relevant comments are already in the attached document. The data presented by the authors are interesting. However, I believe they haven't highlighted the most relevant data. Moreover, the discussion should be revised in a way that it reflects what is shown at the results section. Additionally, the results also does not support the authors conclusions, and limitations should also be revised.

Answer: Suggestions accepted. We have corrected the manuscript, as suggested in the attached document and highlighted the change in the text (yellow).

Reviewer #2: Statistic analysis should be clearly described in the revised manuscript. The authors stated that they had performed comparisons. However, they only reported linear correlation coefficients. Conclusions did not derive from the statistical significance of the data. Moreover, correction for multiple comparisons were not described.

Answer: The statistical analysis was clearly described by a statistic professor (Ghantous, G.F.). We did not make comparisons. This word was removed from the text. We inserted the correlations table (Pearson correlation) inside the text to make it easier to understand. We corrected the conclusion statement.

1) Page 3 line 48 - Correct de reference list.

Answer: Corrected, as suggested. 

2) Page 3 line 56 - Asa 6 does not define anesthetic risk

Answer: Corrected, as suggested. 

3) I believe all this part of the paragraph should be removed 

Answer: The paragraph was removed and the next paragraph was moved to the previous, as suggested.

4) page 4 line 64 - There is no comparison, only description.

Answer: Corrected, as suggested. 

5) Page 15 lince 252 - Show the correlation figures.

Answer: Corrected, as suggested. The supplementary file was inserted in the text.

6) Page 16 line 261 - Is it really the first? This statement looks a little bit presumptuous 

Answer: Corrected, as suggested. 

7) page 16 line 262 - The results does not support this conclusion.

Answer: Corrected, as suggested. The paragraph was removed.

10) Page 17 line 285 - The readers deserve a little bit of explanation here.

Answer: Corrected, as suggested. We added the suggested explanation in the manuscript (why it iso difficult to manage PaCO2 during anesthesia in a recumbent position).

11) page 18 line 329 - Unnecessary explanation here.

Answer: Corrected, as suggested. The paragraph was removed.

12) Page 21 libre 390 - State the limitation imposed by the lack of controls.

Answer: Corrected, as suggested. This limitation was highlighted. 

13) Page 22 line 393 - I believe the conclusion has a lot of room for improvement. The authors should try something like that

Answer: Corrected, as suggested.

---

## [Decision Letter · Decision Letter 1]

28 Oct 2024

Anesthesia for non-obstetric surgery during late term pregnancy in mares

PONE-D-24-15168R1

Dear Dr. Dória,

We’re pleased to inform you that your manuscript has been judged scientifically suitable for publication and will be formally accepted for publication once it meets all outstanding technical requirements.

Kind regards,

Daniel Negrini Medeiros, PhD

Guest Editor

PLOS ONE

Additional Editor Comments (optional):

I believe the authors have thoroughly addressed all the remarks from both reviewers. Reviewer #1, in particular, provided valuable insights on the paper’s structure, and the authors made substantial efforts to incorporate these suggestions. Reviewer #2’s key comments focused on statistical analysis, which the authors also addressed effectively. As a result, I believe the manuscript is now suitable for publication in PLOS ONE.

Reviewers' comments:

Reviewer's Responses to Questions

**Comments to the Author**

1. If the authors have adequately addressed your comments raised in a previous round of review and you feel that this manuscript is now acceptable for publication, you may indicate that here to bypass the “Comments to the Author” section, enter your conflict of interest statement in the “Confidential to Editor” section, and submit your "Accept" recommendation.

Reviewer #1: All comments have been addressed

Reviewer #2: All comments have been addressed

2. Is the manuscript technically sound, and do the data support the conclusions?

Reviewer #1: Yes

Reviewer #2: (No Response)

3. Has the statistical analysis been performed appropriately and rigorously? 

Reviewer #1: Yes

Reviewer #2: (No Response)

4. Have the authors made all data underlying the findings in their manuscript fully available?

Reviewer #1: Yes

Reviewer #2: (No Response)

5. Is the manuscript presented in an intelligible fashion and written in standard English?

Reviewer #1: Yes

Reviewer #2: (No Response)

6. Review Comments to the Author

Reviewer #1: All comments have been properly addressed by the authors. I believe now the manuscript is solid and coherent and reached level for publication in PLOS ONE

Reviewer #2: (No Response)

7. PLOS authors have the option to publish the peer review history of their article (what does this mean?). If published, this will include your full peer review and any attached files.

Reviewer #1: No

Reviewer #2: **Yes: **Sergio Luis Schmidt

---

## [Editor Report · Acceptance letter]

13 Nov 2024

PONE-D-24-15168R1 

PLOS ONE

Dear Dr. Dória, 

I'm pleased to inform you that your manuscript has been deemed suitable for publication in PLOS ONE. Congratulations! Your manuscript is now being handed over to our production team.

Kind regards, 

on behalf of

Professor Daniel Negrini Medeiros 

%CORR_ED_EDITOR_ROLE%

PLOS ONE